# Multimorbidity among Two Million Adults in China

**DOI:** 10.3390/ijerph17103395

**Published:** 2020-05-13

**Authors:** Xiaowen Wang, Shanshan Yao, Mengying Wang, Guiying Cao, Zishuo Chen, Ziting Huang, Yao Wu, Ling Han, Beibei Xu, Yonghua Hu

**Affiliations:** 1Department of Epidemiology and Biostatistics, School of Public Health, Peking University Health Science Center, Beijing 100191, China; wangxw@bjmu.edu.cn (X.W.); yaoshanshan@pku.edu.cn (S.Y.); mywang@bjmu.edu.cn (M.W.); caoguiying@bjmu.edu.cn (G.C.); Chenzishuo@bjmu.edu.cn (Z.C.); huangzt@pku.edu.cn (Z.H.); yaowu@pku.edu.cn (Y.W.); 2Medical Informatics Center, Peking University Health Science Center, Beijing 100191, China; xubeibei@bjmu.edu.cn; 3Department of Medicine, Yale School of Medicine, New Haven, CT 06510, USA; ling.han@yale.edu

**Keywords:** multimorbidity, prevalence, pattern, chronic diseases, China

## Abstract

To explore the multimorbidity prevalence and patterns among middle-aged and older adults from China. Data on thirteen chronic diseases were collected from 2,097,150 participants aged over 45 years between January 1st 2011 and December 31st 2015 from Beijing Medical Claim Data for Employees. Association rule mining and hierarchical cluster analysis were applied to assess multimorbidity patterns. Multimorbidity prevalence was 51.6% and 81.3% in the middle-aged and older groups, respectively. The most prevalent disease pair was that of osteoarthritis and rheumatoid arthritis (OARA) with hypertension (HT) (middle-aged: 22.5%; older: 41.8%). Ischaemic heart disease (IHD), HT, and OARA constituted the most common triad combination (middle-aged: 11.0%; older: 31.2%). Among the middle-aged group, the strongest associations were found in a combination of cerebrovascular disease (CBD), OARA, and HT with IHD in males (lift = 3.49), and CBD, OARA, and COPD with IHD in females (lift = 3.24). Among older patients, glaucoma and cataracts in females (lift = 2.95), and IHD, OARA, and glaucoma combined with cataracts in males (lift = 2.45) were observed. Visual impairment clusters, a mixed cluster of OARA, IHD, COPD, and cardiometabolic clusters were detected. Multimorbidity is prevalent among middle-aged and older Chinese individuals. The observations of multimorbidity patterns have implications for improving preventive care and developing appropriate guidelines for morbidity treatment.

## 1. Introduction

Multimorbidity, commonly defined as the cooccurrence of two or more chronic diseases, has gained increasing attention as a prominent public health issue worldwide [1]. Multimorbidity has been associated with poorer health outcomes including greater risk of disability and mortality among older populations [2]. Additionally, it complicates treatment and health care plans and increases the risk of psychiatric and somatic complications, thus leading to higher healthcare expenditures and resource utilization [3,4].

A large body of literature has reported multimorbidity prevalence and patterns worldwide. It was reported that the prevalence of multimorbidity ranged from 45% to 72% among older adults aged 50 years and over in nine countries [5] and ranged from 24% to 83% among those aged 60 years and older in South Asia [6]. In addition, the overall prevalence of multimorbidity among older adults is widely ranged from 6.4% to 86.9% in China [7]. Additionally, studies have observed different multimorbidity patterns, such as cardiometabolic patterns, mental health patterns, and musculoskeletal patterns, using cluster analysis, factor analysis, or Association rule mining (ARM) [8,9,10]. Nevertheless, multimorbidity patterns are expected to have unique characteristics for different countries because of a variance in factors such as environment, race/ethnicity, and lifestyles [11]. 

However, most research studies investigating multimorbidity in China have been conducted among community-dwelling populations and defined chronic conditions using self-reported questionnaires or face-to-face interviews, which can be affected by recall and selection bias [8,9,12,13,14]. The population-based estimates derived from claims data were very limited, especially in Asian countries [14,15,16]. Studies based on claims data defined multimorbidity using more objective clinical diagnoses without recall/non-response bias based on a patient-based population [17,18]. Furthermore, using International Classification of Diseases (ICD) codes in claims data also facilitated the international comparability of understanding multimorbidity [19].

In addition, current multimorbidity studies have mostly focused on older adults aged ≥60 years [20,21,22,23]. One recent study reported that multimorbidity had a greater impact on all-cause mortality in middle-aged populations than in older populations [24]. There was found to be a greater hospitalization burden among middle-aged people with multimorbidity than their older counterparts as well [25]. Middle-aged patients with multimorbidity represent a vulnerable group with distinct chronic care needs [25]. Studies on multimorbidity among middle-aged adults are therefore essential to address their need for personalized health care. 

Better understanding of multimorbidity prevalence and patterns among different populations can help explore possible interactions between different diseases as well as improve population-specific approaches for managing multimorbidity to lessen the public health burden [26,27]. This study therefore aimed to evaluate the prevalence and patterns of multimorbidity (i.e., combinations and clustering of chronic diseases) among middle-aged and older Chinese adults using claims data accumulated through a mandatory insurance programme in Beijing, China. 

## 2. Materials and Methods

### 2.1. Data Source

This study used data from the Beijing Medical Claim Data for Employees (BMCDE), a mandatory health insurance programme consisting of all working and retired workers in Beijing [28,29]. BMCDE included information on patients’ demographic characteristics (age and sex), inpatient and outpatient clinical diagnoses, and medications, with personal identifiers removed. All clinical visits of patients could be linked with their unique encrypted identification number. This study was considered exempt because BMCDE data were collected for administrative purposes with personal identifiers removed for research purposes.

### 2.2. Measurement of Multimorbidity and Study Population

For this study, the following thirteen chronic diseases were selected based on the most frequently mentioned diseases for multimorbidity measures by previous literatures that were considered to significantly impact long-term treatment and decrease functional performance and quality of life among the Chinese population [30,31]: malignancy, cerebrovascular disease (CBD), ischaemic heart disease (IHD), chronic obstructive pulmonary disease (COPD), diabetes mellitus (DM), depressive disorders (DD), chronic kidney disease (CKD), osteoarthritis and rheumatoid arthritis (OARA), peptic ulcer disease (PUD), cataracts, heart failure (HF), hypertension (HT), and glaucoma. Diseases were identified if they had been documented using inpatient or outpatient ICD-10 codes at least twice in individual medical records during the 5-year period from January 1, 2011 to December 31, 2015 [32,33]. For this study, a total of 2,097,150 people aged ≥45 years in 2015 were included. Appendix A lists all chronic medical diseases included and their corresponding ICD-10 codes. For this study, multimorbidity was defined as concurrently suffering from two or more chronic diseases.

### 2.3. Statistical Analyses

Age was categorized into the following two subgroups: 45–59 years (middle-aged group) and ≥60 years (older group). Continuous variables were examined using the mean (standard deviation, SD). Categorical variables were summarized with frequencies (percentages), and the *X*^2^ test was performed to compare these characteristics. Given the sufficiently large sample size in this study, all tests showed a significant p value (*p* < 0.001), which is not presented in the results [34].

ARM was applied to determine common multimorbidity patterns that met a minimum requirement of measurement indicators among large sets of diagnoses. Association rules were relationships between sets of diseases from “antecedent” to “consequent” [35]. The following are the three common measurement indicators in ARM: support (the frequency of the disease combinations in the association rule), confidence (how frequently the antecedent diseases occur, conditional on the consequent diseases), and lift (how much more frequently the diseases set in the same association rule occur together compared with the expected prevalence under statistical independence). Lift was considered the main measure of significance in ARM [36]. The higher the lift, the higher the chance of co-occurrence of the consequent with the antecedent and the more significant the association [36]. For this study, the minimum thresholds of parameters in ARM were defined as follows: support > 2%, confidence > 10%, and lift > 1%. Hierarchical cluster analysis (HCA) was used to identify those diseases that were inclined to gather together according to their correlated structure and similarity patterns. Those that were strongly correlated or had similar diseases were classified into the same clusters using homogeneity criteria [37]. The homogeneity criterion of a cluster is defined as the sum of correlation ratios [37]. The function “hclustvar” in the R package “ClustOfVar” was used in the clustering analysis (www.r-project.org). All analyses were performed using R 3.2.2 (R Development Core Team).

## 3. Results

Table 1 shows the basic demographics and prevalence of chronic diseases in the studied population. This study included 2,097,150 people aged ≥45 years, with 50.0% (*n* = 1,048,575) aged 45–59 years (middle-aged group) and 50.0% (*n* = 1,048,575) aged ≥60 years (older group). The mean age was 52.8 years for those aged 45–59 years and 70.3 years for those aged ≥60 years.

Figure 1 shows the multimorbidity prevalence by age and sex. The prevalence of multimorbidity was 51.6% (male: 46.1%; female: 55.7%) in the middle-aged group and 81.3% (male: 78.8%; female: 83.7%) in the older group. The overall proportion of people with multimorbidity increased from 51.6% for those aged 45–59 years to 89.3% for those over 90 years. Females had a higher prevalence than males for each age group.

The most frequent (top 3) combinations, stratified by age, sex, and total number of chronic diseases, are presented in Table 2. Among patients with at least two chronic conditions, OARA combined with HT was the most prevalent disease pair for patients of all ages (male: 29.6%; female: 37.6%), male middle-aged patients (male: 16.0%; female: 27.3%), and male older patients (35.1%). Among older females, IHD combined with OARA was the most prevalent disease pair (49.2%). Furthermore, IHD, HT, and OARA was the most common triad combination among patients with at least three chronic conditions (middle-aged male: 8.0%; middle-aged female: 13.2%; older male: 26.4%; older female: 36.6%).

The ARM results illustrated the associations between these diseases. The top 10 association rules among the included diseases according to lift are shown in Table 3. Among the middle-aged patients, the strongest association was found between the combination of CBD, OARA, HT, and IHD in males where the lift was 3.49, whereas the highest lift for females was 3.24 for the combination of CBD, OARA, COPD, and IHD, making these diseases occur together with 3.49- or 3.24-times higher likelihoods, respectively, than would be expected if they were independent. Among older adults, the strongest associations were observed between glaucoma and cataracts in females (lift = 2.952) and the combination of IHD, OARA, and glaucoma with cataracts in males (lift = 2.450), indicating that these antecedent combinations positively correlated with the occurrence of cataracts. Moreover, in the middle-aged group, IHD tended to be comorbid with others that occurred in 9 association rules in males and 10 in females among the top 10 rules. In contrast, among the older males, cataracts and diabetes were the most common comorbidities, and both occurred in 5 association rules. 

Disease clusters derived from HCA stratified by age and sex are shown in Figure 2a–d. Common disease clusters observed in the four subgroups included a visual impairment cluster consisting of glaucoma and cataract and a mixed cluster of OARA, IHD, and COPD. Furthermore, a cardiometabolic cluster, including HF, HT, DM and CKD, was detected among middle-aged female individuals and the older group.

## 4. Discussion

To the best of our knowledge, this study was among the first to explore multimorbidity prevalence and patterns based on large medical claims data among a middle-aged and older Chinese population. The prevalence of multimorbidity increased with age. Females had a higher prevalence of multimorbidity than males. OARA, combined with HT, was the most prevalent disease pair for patients of all ages. IHD, HT, and OARA was the most common triad combination. In addition, several disease clusters were identified in our study including a visual impairment cluster, a mixed cluster of OARA, IHD, and COPD, and a cardiometabolic cluster. There were strong associations between CBD, OARA, and IHD among the middle-aged group and between glaucoma and cataracts among the older group. 

We found that the prevalence of multimorbidity was 51.6% for middle-aged adults aged 45–59 and 81.3% for older adults aged ≥60 years. One study using claims data from Taiwan in 2013 showed a similar prevalence of multimorbidity, which was 56.8%, 74.6%, and 82.6% in people aged 60–69, 70–79, and 80–89 years, respectively [16]. Additionally, two European studies based on claims data also reported that the multimorbidity prevalence among adults aged ≥60 years old was 85% in Germany [14] and 77% in Switzerland [15]. Consistent with previous studies, we observed a higher prevalence of multimorbidity among females than among males, which may be partially due to the relatively higher prevalence of select chronic diseases in females [8,38,39,40]. One study conducted in six low- and middle-income countries among adults aged ≥50 years reported a higher overall prevalence of six chronic diseases, including angina, arthritis, asthma, chronic lung disease, depression, and HT, in females [41]. Moreover, the incidence of chronic inflammatory diseases in females after menopausal age, as a result of hormonal changes, exceeded that observed in males [42]. One alternative explanation might be that females were generally more aware of their own health status and had more healthcare-seeking behaviours, which might lead to a higher probability of being diagnosed with diseases [43].

According to the frequency, we found that the most prevalent disease pair was OARA combined with HT, and the most common triad combination was IHD, HT, and OARA. A multi-country study reported similar results: comorbid diseases of HT and arthritis were highly prevalent in China, Finland, Ghana, Poland, Russia, South Africa, and Spain [5]. A study in Belgium also detected the most prevalent pair as HT-OA [44]. Moreover, the lift in ARM discovered interesting combinations of comorbidities that were occurring more frequently than expected. In our study, the ranked lift indicated that there were strong associations between vascular and musculoskeletal disorders, including CBD, IHD, and HT with OARA. A cross-sectional study conducted in Europe observed that HT was one of the most common comorbidities among individuals with osteoarthritis (OA) [45]. In addition, a meta-analysis showed that there was a significant relationship between cardiovascular disease (CVD) and OA [46,47]. Possible explanations for the relationship between CVD and OARA include shared biological mechanisms, such as ageing and obesity [48,49,50]. For example, obesity can induce chronic inflammation and endothelial dysfunction, which are simultaneously implicated in cartilage degradation, atherosclerosis, and a higher CVD risk [51]. Overweight and obese people also have reduced levels of physical activity and functional limitations, which are concomitant risk factors for CVD, joint pain and OARA [52]. Moreover, individuals with nonsteroidal anti-inflammatory medication treatments could contribute substantially to the OA–CVD association [53]. The observations of prevalent disease combinations could have implications for health prevention, where measurements targeting a specific factor may benefit two or more related diseases.

Furthermore, strong associations between IHD, OARA, and COPD were also detected by ARM and HCA. Our findings were consistent with those from a study using medical record data in Sweden which reported significantly associated comorbidities, including IHD and OA, among COPD patients [54]. Similarly, a population-based cohort in Canada also observed COPD and HT as prevalent comorbidities among people with OA [55]. Moreover, a systematic review confirmed that the prevalence of OA was high among individuals with COPD [56]. The potential mechanisms might include increased systemic inflammatory mediators and some adverse effects, such as physical inactivity, which are also risk factors for other diseases [56]. Compared to a study conducted in rural China, it observed the most prevalent multimorbidity pattern for CBD, HT, and DM, which might be attributed to an imbalance in the delivery of primary care services, such as the lack of blood pressure and blood glucose monitoring, and availability of essential medicines in rural districts [57]. Moreover, in older females, strong associations between IHD, OARA, glaucoma, and cataracts were observed, partially owing to the high prevalence of these diseases [58]. The ARM results also reported common comorbidities, including IHD among middle-aged individuals and DM in males. Survival bias may explain the age and gender difference for the comorbidity, as IHD and DM were linked to higher mortality in older [59] and female [60] patients, respectively.

The main strength of this study was the large sample size of the administrative data used to illustrate multimorbidity prevalence and patterns among a middle-aged and older Chinese population. The multimorbidity measures defined by the ICD codes, including a broad range of chronic diseases, can help detect specific patterns potentially left out in survey-based studies. This study also had several limitations. First, the BMCDE database only included people who had been employed, leading to limited generalizability to a general population. Additionally, although claims data included ICD-10 codes recorded by well-trained doctors in certified hospitals, we could not necessarily ensure exclusion of inaccurate or underreporting diagnoses. Finally, the dataset did not include patients’ socioeconomic status, anthropometric parameters, detailed clinical information, as well as health-related behaviours. We therefore could not investigate potential confounding factors and take disease severity into consideration.

## 5. Conclusions

Multimorbidity was prevalent among middle-aged and older Chinese individuals. Females had a higher prevalence than males. The most prevalent disease pair was OARA combined with HT. The most common triad combination consisted of IHD, HT, and OARA. CVDs (including CBD, IHD, and HT), OARA and COPD were more likely to co-occur. The observations of the multimorbidity pattern, including the most frequent comorbidities, associations, and clusters in our study, provide evidence for the priority of disease clusters for further research and encourage healthcare providers to develop healthcare plans towards a multiple-condition framework among patients with multimorbidity. Deep understanding of the complexity of multimorbidity based on large administrative datasets can improve the quality and efficiency of patient-centred care.

## Figures and Tables

**Figure 1 ijerph-17-03395-f001:**
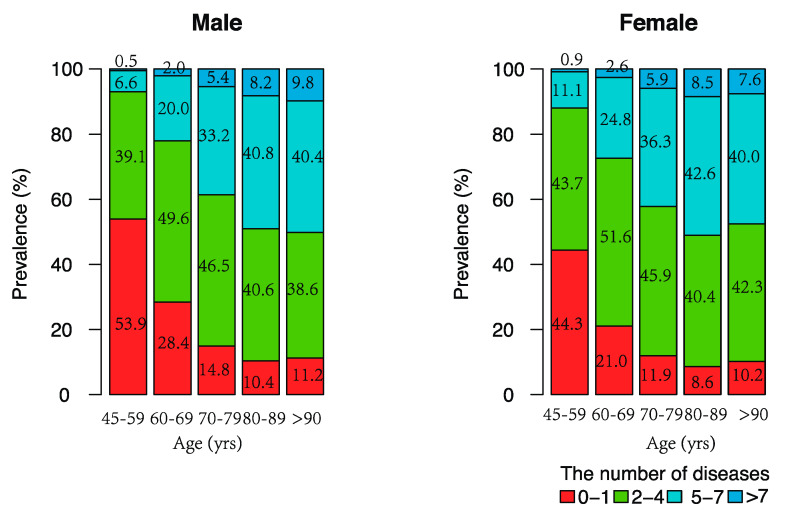
Multimorbidity prevalence by age and sex among middle-aged and older Chinese.

**Figure 2 ijerph-17-03395-f002:**
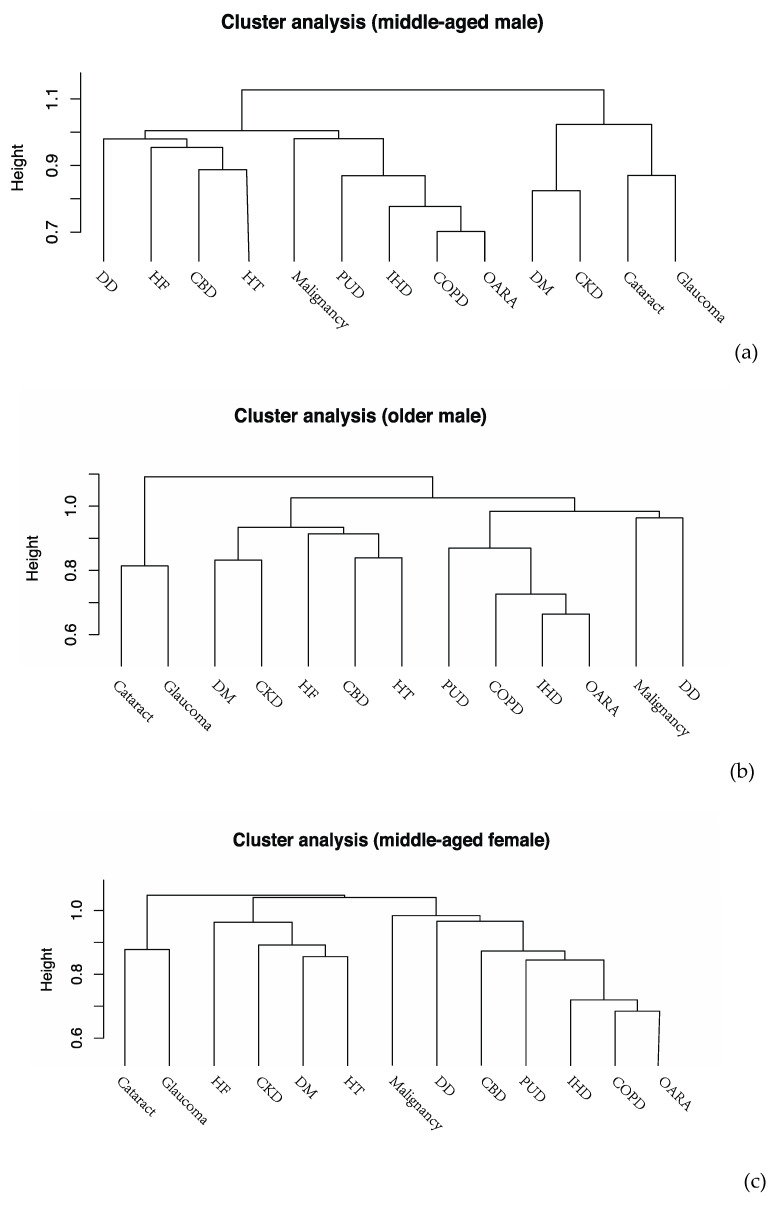
Dendrograms of cluster analysis showing the distribution and aggregation of chronic diseases in (**a**) middle-aged males, (**b**) middle-aged females, (**c**) older males, and (**d**) older females.

**Table 1 ijerph-17-03395-t001:** Observed prevalence of chronic diseases by age and sex.

	45 ≤ Age ≤ 59			Age ≥ 60		
	All (*n* = 1,048,575)	Male (*n* = 446,540)	Female (*n* = 602,035)	All (*n* = 1,048,575)	Male (*n* = 508,717)	Female (*n* = 539,858)
**Age (yrs), mean (SD)**	52.8 (4.3)	52.7 (4.3)	52.9 (4.2)	70.3 (8.0)	70.5 (8.1)	70.0 (8.0)
**Chronic health disease, *n* (%)**						
**Malignancy**	58,077 (5.5)	28,112 (6.3)	29,965 (5.0)	94,008 (9.0)	44,195 (8.7)	49,813 (9.2)
**CBD**	81,777 (7.8)	39,936 (8.9)	41,841 (6.9)	263,451 (25.1)	138,135 (27.2)	125,316 (23.2)
**IHD**	231,483 (22.1)	112,274 (25.1)	119,209 (19.8)	587,596 (56.0)	273,645 (53.8)	313,951 (58.2)
**COPD**	206,724 (19.7)	100,081 (22.4)	106,643 (17.7)	394,699 (37.6)	181,243 (35.6)	213,456 (39.5)
**DM**	216,014 (20.6)	104,721 (23.5)	111,293 (18.5)	317,576 (30.3)	163,205 (32.1)	154,371 (28.6)
**DD**	113,530 (10.8)	55,000 (12.3)	58,530 (9.7)	128,534 (12.3)	51,017 (10.0)	77,517 (14.4)
**CKD**	57,640 (5.5)	27,944 (6.3)	29,696 (4.9)	130,281 (12.4)	66,252 (13.0)	64,029 (11.9)
**OARA**	467,787 (44.6)	226,943 (50.8)	240,844 (40.0)	707,436 (67.5)	306,174 (60.2)	410,262 (74.3)
**PUD**	125,720 (12.0)	61,089 (13.7)	64,631 (10.7)	190,506 (18.2)	89,638 (17.6)	100,868 (18.7)
**Cataract**	37,363 (3.6)	18,151 (4.1)	19,212 (3.2)	216,475 (20.6)	88,028 (17.3)	128,447 (23.8)
**HF**	10,835 (1.0)	5280 (1.2)	5555 (0.9)	48,672 (4.6)	24,717 (4.9)	23,955 (4.4)
**HT**	520,324 (49.6)	252,602 (56.6)	267,722 (44.5)	681,179 (65.0)	335,584 (66.0)	345,595 (64.0)
**Glaucoma**	23,188 (2.2)	11,212 (2.5)	11,976 (2.0)	51,695 (4.9)	21,195 (4.2)	30,500 (5.6)

Abbreviations: SD, standard deviation; CBD, cerebrovascular disease; IHD, ischaemic heart disease; COPD, chronic obstructive pulmonary disease; DM, diabetes mellitus; DD, depressive disorders; CKD, chronic kidney disease; OARA, osteoarthritis and rheumatoid arthritis; PUD, peptic ulcer disease; HF, heart failure; HT, hypertension.

**Table 2 ijerph-17-03395-t002:** Top three frequent unique combination clusters for patients with multimorbidity, stratified by total number of chronic diseases and by sex.

Age Group	Rank	Dyads of Morbidities				Triads of Morbidities			
		Male		Female		Male		Female	
		Combination	*n* (%)	Combination	*n* (%)	Combination	*n* (%)	Combination	*n* (%)
All ages; male: *n* = 955,257, female: *n* = 1,141,893									
	1	OARA + HT	282,995 (29.6)	OARA + HT	429,553 (37.6)	IHD + OARA + HT	194,735 (20.4)	IHD + OARA + HT	276,928 (24.3)
	2	IHD + HT	272,230 (28.5)	IHD + OARA	383,889 (33.6)	COPD + OARA + HT	132,923 (13.9)	COPD + OARA + HT	204,352 (17.9)
	3	IHD + OARA	255,872 (26.8)	IHD + HT	324,411 (28.4)	IHD + COPD + OARA	130,557 (13.7)	IHD + COPD + OARA	198,311 (17.4)
45–59 years; male: *n* = 446,540, female: *n* = 602,035									
	1	OARA + HT	71,626 (16.0)	OARA + HT	164,234 (27.3)	IHD + OARA + HT	35,719 (8.0)	IHD + OARA + HT	79,290 (13.2)
	2	DM + HT	65,518 (14.7)	COPD + OARA	120,912 (20.1)	DM + OARA + HT	24,719 (5.5)	COPD + OARA + HT	70,249 (11.7)
	3	IHD + HT	63,979 (14.3)	IHD + OARA	118,355 (19.7)	COPD + OARA + HT	24,367 (5.5)	IHD + COPD + OARA	58,021 (9.6)
≥60 years; male: *n* = 602,035, female: *n* = 539,858									
	1	OARA + HT	211,369 (35.1)	IHD + OARA	265,534 (49.2)	IHD + OARA + HT	159,016 (26.4)	IHD + OARA + HT	197,638 (36.6)
	2	IHD + HT	208,251 (34.6)	OARA + HT	265,319 (49.1)	IHD + COPD + OARA	110,904 (18.4)	IHD + COPD + OARA	140,290 (26.0)
	3	IHD + OARA	206,397 (34.3)	IHD + HT	230,795 (42.8)	COPD + OARA + HT	108,556 (18.0)	COPD + OARA + HT	134,103 (24.8)

Abbreviations: IHD, ischaemic heart disease; COPD, chronic obstructive pulmonary disease; DM, diabetes mellitus; OARA, osteoarthritis and rheumatoid arthritis; HT, hypertension.

**Table 3 ijerph-17-03395-t003:** The top 10 association rules in the order of lift.

Middle-Aged						Older					
Male			Female			Male			Female		
Antecedent	Consequent	Lift	Antecedent	Consequent	Lift	Antecedent	Consequent	Lift	Antecedent	Consequent	Lift
CBD + OARA + HT	IHD	3.487	CBD + OARA + COPD	IHD	3.237	IHD + OARA + Glaucoma	Cataract	2.450	Glaucoma	Cataract	2.952
CBD + OARA	IHD	3.321	CBD + COPD	IHD	3.169	IHD + Glaucoma	Cataract	2.419	OARA + PUD + CKD	COPD	2.008
OARA + HT + PUD	IHD	3.020	CBD + OARA + HT	IHD	3.134	HT + Glaucoma	Cataract	2.417	IHD + DD + OARA + HT	IHD	2.006
COPD + OARA + HT	IHD	2.981	CBD + OARA	IHD	2.992	OARA + Glaucoma	Cataract	2.382	IHD + COPD + DD	IHD	1.997
DM + OARA + HT	IHD	2.945	CBD + HT	IHD	2.874	Glaucoma	Cataract	2.311	IHD + CKD + PUD	COPD	1.992
COPD + DM	IHD	2.781	COPD + DM + OARA + HT	IHD	2.821	CBD + CKD + HT	DM	2.062	CKD + PUD + HT	COPD	1.961
COPD + HT	IHD	2.692	COPD + PUD + OARA + HT	IHD	2.764	IHD + CKD + HT	DM	1.953	IHD + DM + OARA + PUD + HT	IHD	1.957
PUD + HT	IHD	2.637	COPD + DM + HT	IHD	2.736	CBD + IHD + CKD	DM	1.946	CBD + IHD + CKD + OARA + HT	COPD	1.956
COPD + PUD	OARA	2.623	COPD + PUD + HT	IHD	2.711	IHD + CKD + OARA + HT	DM	1.937	IHD + DD + HT	IHD	1.952
CBD + HT	IHD	2.593	CBD	IHD	2.692	CBD + CKD	DM	1.933	CKD + OARA + Cataract	COPD	1.945

Abbreviations: CBD, cerebrovascular disease; IHD, ischaemic heart disease; COPD, chronic obstructive pulmonary disease; DM, diabetes mellitus; DD, depressive disorders; CKD, chronic kidney disease; OARA, osteoarthritis and rheumatoid arthritis; PUD, peptic ulcer disease; HT, hypertension.

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
