# Peer review of "Multimorbidity among Two Million Adults in China"

_ijerph, 2020, doi:10.3390/ijerph17103395_

Round 1

Reviewer 1 Report

This is an interesting paper with creative approach applied on a big data set. However, there some fundamental questions on their findings need to be clarified.   

  1. In their conclusion, one of the major finding is that OARA+HT is the most prevalent disease pairs, and IHD+HT+OARA were the most common triad combinations.

Since OARA and HT were the more prevailing morbidities for all age-gender categories (Table 1), naturally by randomly distributing, people will be most likely to have OARA-HT pair than other pairs. For example in table 1, 50.8% and 56.6% of mid-age male had OARA and HT respectively, then by chance there will be 28.8% (0.508*0.566) having OARA-HT pair, which is close to the actual prevalence 29.6% in Table 2.

So, is there any implication of this most prevailing Dyad of morbidities? Any of the prevalence is significantly high/low in statistical sense?

Likewise, IHD, HT and OARA are the top 3 prevailing morbidities, is there anything surprising to observe these threes as the most prevailing triad?

  1. By using aggregate clustering and dendrogram, the authors observed interesting combinations like CBD+IHD+HT, and OARA+COPD. Are their prevalence actually significantly higher than by chance?  

Reviewer 2 Report

The manuscript explores the multimorbidity prevalence and patterns among middle-aged and older adults from China, based on the database of Beijing Medical Claim, from 2011 to 2015.

The main results of the study were that the multimorbidity was prevalent among middle-aged (45-59 years) and older Chinese individuals (>60 years). Females had a higher prevalence than males. Osteoarthritis and rheumatoid arthritis and Hypertension were the most prevalent disease pairs. Ischaemic heart disease, Hypertension, and osteoarthritis and rheumatoid arthritis were the most common triad combinations.

 The manuscript is well structured and the conclusions are in accordance with the results. To improve the quality of paper before publication I suggest a  minor revision as indicate in these points:

  • in the introduction it’s necessary better clarify the purpose and the main goal of the study

  • an important point, not discussed by the authors, it’s the possible correlations of disease with anthropometric parameters, in particular the body weight and BMI. The presence of overweight and obesity, is correlated with the onset of different disease, such as hypertension, cerebrovascular (Ricci et al., Clin Exp Hypertes 2017; 39: 8-16) and cardiovascular disease (Ortega et al., Circ Res. 2016;118:1752-70) in particular in older population;

  • a discussion of the results, based on the educational level, socio-economical and occupational status of the population studied, could be interest to identify possible correlations with the disease and comorbidity;

Minor points:

  • In the manuscript it’s necessary an English revision; moreover different typing errors are present;

  • in Table 2 I suggest to clarify the numbers in the first column;

  • at the end of paper, a list of abbreviations could be useful the read in a easy way the manuscript.

Round 2

Reviewer 1 Report

The authors had sufficiently answered my questions.